# Impact of Vaccinating Adult Women Who Are HPV-Positive or with Confirmed Cervical SIL with the 9-Valent Vaccine—A Systematic Review

**DOI:** 10.3390/v17101377

**Published:** 2025-10-15

**Authors:** Dominik Pruski, Sonja Millert-Kalińska, Robert Jach, Jakub Żurawski, Marcin Przybylski

**Affiliations:** 1Department of Obstetrics and Gynecology, District Public Hospital, Juraszów 7-19, 60-479 Poznań, Poland; millertsonja@gmail.com (S.M.-K.); nicramp@poczta.onet.pl (M.P.); 2Division of Gynecologic Endocrinology, Jagiellonian University Medical College, Kopernika 23, 31-501 Krakow, Poland; jach@cm-uj.krakow.pl; 3Department of Immunobiology, Poznan University of Medical Sciences, 60-512 Poznań, Poland; zurawski@ump.edu.pl

**Keywords:** HPV vaccination in HPV-positive adults, HPV vaccination in adults, 9-valent vaccine, HPV vaccination in SIL, conization and vaccination, remission of HPV after vaccination

## Abstract

Infection with oncogenic human papillomavirus (HPV) remains a leading cause of cervical cancer and other HPV-related diseases. This situation persists despite the availability of effective prophylactic vaccines. While global vaccination programs have significantly reduced the incidence of HPV in adolescents and young adults, many women presenting with HPV infection or squamous intraepithelial lesions (SIL) were not covered by primary prevention. This review was performed with the aim of evaluating the impact of administering the 9-valent HPV vaccine in adult women who are HPV-positive or have histologically confirmed cervical precancerous lesions. Following the PRISMA 2020 guidelines, a search was performed in the MEDLINE, Scopus, and Cochrane Library databases. A total of 653 studies were retrieved, of which 7 studies, including 19,414 women, met the inclusion criteria. According to the literature, vaccination was linked to significant reductions in persistent HPV infection, progression of SIL, and recurrence of high-grade lesions after surgical removal. Complete HPV remission was achieved in up to 72.4% of vaccinated women, compared to 45.7% among unvaccinated controls. Vaccination after conization lowered the recurrence risk of CIN2+ lesions by 87%, with benefits seen regardless of timing. The most significant effect was observed when vaccine administration was performed before the surgical procedure. Furthermore, HPV vaccination notably enhanced viral clearance and decreased the likelihood of repeated surgical interventions. Despite differences in study design and follow-up definitions, the overall evidence supports additional vaccination in HPV-positive adult women as an effective measure to reduce recurrence and promote viral remission. These findings emphasize the need for clear guidelines and wider access to HPV vaccination for adult populations.

## 1. Introduction

Infection with oncogenic HPV types—despite being entirely preventable—remains a common cause of anogenital, head, and neck diseases, including warts, precancerous lesions, and cancer [1,2]. The current World Health Organization (WHO) has launched a global strategy to eliminate cervical cancer by 2030 through a combination of vaccination, screening, and treatment. The strategy aims to achieve 90% vaccination coverage for girls under 15, testing of 70% of women aged 35 to 45, and 90% of women diagnosed with cervical cancer receiving treatment [3]. In countries such as Australia, New Zealand, Scandinavia, and the United Kingdom, where national HPV vaccination programs were introduced in the 1980s, infection rates have already declined markedly following the implementation of bivalent and later quadrivalent prophylactic vaccines.

By contrast, in countries where HPV vaccination was introduced decades later, precancerous lesions and cervical cancers remain prevalent. Doctors often recommend additional HPV vaccinations to women who are not covered by preventive vaccinations, in addition to surgical treatment. However, there are no clearly defined schedules, timing of the first dose in relation to the time of surgery, or recommendations for vaccination valency. At present, the most widely available vaccine is Gardasil 9, which covers seven oncogenic HPV types and two low-oncogenic types that cause warts [4].

Access to adult HPV vaccination is inconsistent worldwide. Only a few European countries offer free vaccination with the 9-valent vaccine for adults; for example, Sweden provides vaccination for all interested university students, and Germany provides vaccination to MSM and immunosuppressed individuals. Other countries allow vaccination for adults, but it is typically not reimbursed. The vaccination policy in adults is not uniform and varies across the globe. The U.S. Centers for Disease Control and Prevention (CDC) strongly recommends HPV vaccination for children before sexual initiation, but there are no data recommending vaccination for adults. For adults aged 27 to 45, the CDC and the Advisory Committee on Immunisation Practices (ACIP) do not routinely recommend HPV vaccination; instead, shared clinical decision-making is advised, meaning that vaccination may be considered in individuals who may benefit. In Asian countries, vaccination for adults is also not systematic. In an integrative literature review covering HPV vaccination recommendations in the Asia-Pacific region, many national immunization programs focus primarily on adolescent girls; few offer adult catch-up or adult vaccination policies. However, there are some individual countries that have catch-up schedules or extended-age policies [5,6,7].

To address this issue, we conducted a systematic literature review (SLR) in three electronic databases (Medline [via PubMed], Elsevier, and Cochrane), using a comprehensive set of search terms, on 9 July 2025. The primary objective of this work is to assess the impact of the 9-valent HPV vaccine in adult women who are HPV-positive or have confirmed cervical SIL. At present, there is no standardized algorithm for HPV vaccination in the case of HPV-positive women, those with LSIL-CIN 1 lesions, or those undergoing treatment of precancerous cervical conditions such as HSIL-CIN 2 and CIN 3. The secondary objective is to evaluate the role of vaccination in reducing recurrence after excisional treatment and assess whether the timing of vaccination influences outcomes.

## 2. Materials and Methods

### 2.1. Literature Search

We conducted this review using three databases—PubMed, Scopus, and Cochrane Library—in accordance with the PRISMA 2020 guidelines [8], followed by a manual search. The review protocol was registered in PROSPERO (ID: 1157956). The search combined controlled vocabulary (MeSH terms) and free-text terms related to HPV vaccination, HPV infection, cervical intraepithelial lesions, and surgical management. Boolean operators (“AND”, “OR”) and truncations were applied as appropriate. The details of the search design are shown in Table 1. All databases were last accessed on 9 July 2025.

Two authors of this study screened the titles and abstracts of the retrieved studies independently. In the next step, the full texts of 25 studies were assessed by 2 other authors to determine whether they satisfied the inclusion criteria. Table 2 presents the inclusion and exclusion criteria, the PRISMA flowchart is shown in Figure 1, and the division into retrospective and prospective studies is shown in Figure 2. The search strategy according to the PICO framework is presented in Appendix A. We did not need to contact the authors of the included studies to ask for any additional information.

### 2.2. Data Extraction

In the next step, we designed an initial data extraction form and checked its relevance on three randomly chosen studies. Two researchers independently assessed its suitability, and the information from each study was obtained separately by two researchers. In cases of disagreement, conflicts were resolved through discussion. Primarily influenced by the various study designs, as well as the adopted definitions and measurement scales, we did not perform any statistical synthesis of the results.

### 2.3. Risk-of-Bias Assessment

Two independent reviewers (D.P. and S.M.-K.) assessed the risk of bias using the Newcastle–Ottawa quality assessment scale (NOS) for cohort studies [16]. The NOS was applied to evaluate the quality and potential risk of bias of studies across three domains: participant selection, comparability of study groups, and outcome assessment. Discrepancies in scoring were resolved through discussion until consensus was reached between the two independent reviewers. Studies were classified according to the total NOS scores as follows: 0–3 (unsatisfactory), 4–5 (satisfactory), 6–7 (good), and 8–9 (very good) (Table 3).

## 3. Results

### 3.1. Characteristics of the Included Studies

The implemented literature search identified 653 articles. After excluding duplicates, reviews, meta-analyses, book chapters, case reports, and letters, and following a manual search, seven studies met the inclusion criteria. Details on the selection process are presented in the customized PRISMA flowchart shown in Figure 1, while the primary characteristics of the studies included in this review are summarized in Table 3.

The NOS scores (Table 3) in all included studies ranged from 6 to 9 points, indicating a favorable level of methodological quality. All studies demonstrated a strong commitment to addressing the critical aspects of study design and execution, resulting in findings with a high degree of confidence. To evaluate the included studies for quality of evidence and strength of recommendations, we followed the GRADE approach.

These 7 publications examined the impact of the 9-valent vaccine in patients with HPV infection or precancerous cervical lesions, involving a total of 19,414 individuals. Among them, 6437 (33.2%) were vaccinated, while 66.8% (12,977/19,414) were unvaccinated. Two studies were prospective (Krog L. et al. [13] and Pruski D. et al. [14]), with the rest being retrospective analyses. Dvořák V. [9], Palumbo M. [10], Petráš M. et al. [11], Del Pino M. [12] and Henere C. [15] focused on whether HPV vaccination could lower the risk of recurrence following cervical excision forHSIL. One study, conducted by Pruski D. et al. [14], aimed to evaluate HPV resolution post-immunization in initially positive patients. Finally, Krog L. [13], along with their co-authors, investigated whether women vaccinated against HPV and under active surveillance for CIN2 were less likely to progress to CIN3 or worse compared to unvaccinated women.

### 3.2. Definition of HPV Remission/SIL Regression

Regarding the definition of the vaccination effect, different authors have approached the concept of decay/regression in various ways. Polish authors were the only ones to determine the scale of HPV disappearance depending on the HPV genotypes assessed at the time of diagnosis and at least 6–8 months after the last dose of the HPV vaccine, including (1) remains of the same HPV genotype, (2) partial remission of HPV genotypes, (3) complete remission, (4) appearance of an infection with a new genotype, or (5) disappearance of some genotypes and appearance of new ones [14]. Palumbo et al. also presented HPV test results at two follow-up intervals: 6 months and 15 months [10]; however, no information was provided on the number of genotypes included in the study. Petráš M. presented the most extended follow-up. To evaluate the early-, medium-, and long-term effects of HPV vaccination, follow-up periods of 6 months, 18 months, 3 years, and 6 years after cervical excision were analyzed [11]. The maximum observation time was 15 years. In the study by Dvořák V., the disappearance of HPV infection was not assessed; however, the risk of a histopathologically confirmed precancerous condition of the cervix after a LEEP procedure and HPV vaccination was evaluated during a follow-up period of at least 6 months [9]. Krog L. and co-authors carried out a similar observation scheme, but they assessed the presence of precancerous conditions after 28 months [13]. Del Pino M. used three definitions for follow-up. The clinical outcomes of the patients at the end of the follow-up were categorized as follows: (1) persistent/recurrent HSIL (presence of histologically confirmed HSIL/CIN2-3, or a repeat HSIL result in at least two Pap smears separated by six months and a positive HPV testing result, regardless of the histological diagnosis); (2) persistent/recurrent LSIL/HPV (persistent abnormal cytological result of LSIL, ASC-US or AGUS, a single cytology result of HSIL and/or a positive HPV test result without histological diagnosis of HSIL/CIN2-3); and (3) no disease (negative HPV test; negative Pap test; and, if available, a negative biopsy) [12]. The definition of follow-up was the period from conization to either the diagnosis of persistent SIL or the last visit. In the study conducted by Henere C. et al., the clinical outcomes of the patients were categorized as follows: (1) post-treatment HSIL, (2) post-treatment HPV, and (3) no evidence of disease [15].

### 3.3. Impacts of HPV Vaccination on HPV Remission/SIL Regression

The results of all the studies were similar and indicated the benefits of vaccinating adult women after sexual initiation, with HPV infection, or already with cervical disease (Table 4). Italian authors showed that, among women treated for LSIL, a positive HPV test result was obtained in 38% of unvaccinated women compared to 18% in vaccinated women (*p* = 0.0169). Similarly, a spectacular effect was observed in the group of women treated for HSIL: 18% of unvaccinated women had a positive HPV test result, compared to 8% in the vaccinated group (*p* = 0.0353). Staying with early lesions—i.e., the HPV infection itself—researchers from Poland obtained results showing that the rate of HPV disappearance was significantly higher in the vaccinated group over the follow-up period, when compared to the control group. This applied especially in the case of complete remission—i.e., the complete disappearance of the HPV virus—which was observed in nearly three-quarters of vaccinated women (72.4%), compared to less than half of unvaccinated patients (45.7%). This effect was evident when analyzing the disappearance of HPV genotypes covered by the 9-valent vaccine. The proportion of favorable HPV Gardasil genotype after vaccination/observation was significantly higher in the group with a favorable HPV Gardasil genotype before vaccination/observation (51.9% vs. 0.0%, *p* = 0.001) [14]. Researchers have further observed the effect of the 9-valent vaccine on reducing the risk of progression. Dvořák V. et al. [9] proved that the incidence rate of repeat conization was 18 per 100,000 person-days in the unvaccinated cohort, compared to 2 per 100,000 person-days in the vaccinated group. Del Pino M. [12] achieved a similar result: persistent/recurrent HSIL was less frequent in vaccinated than non-vaccinated women (3.3% vs. 10.7%, *p* = 0.015), and HPV vaccination was associated with a reduced risk of persistent/recurrent HSIL (OR 0.2, 95%CI: 0.1–0.7, *p* = 0.010). Notably, vaccination compliance increased when the vaccine was publicly funded (from 35.9% to 79.1%, *p* < 0.001) [9]. Other authors have compared the different effects of vaccinations administered to various patients. According to Petráš et al. [11], CIN2+ recurrence was observed in 513 unvaccinated women, in 14 prophylactically vaccinated women, and 15 in women vaccinated post-excision. The recurrence within 6 months of conization was reduced by 80% with prophylactic vaccination and by 89% with incomplete post-excision vaccination. Among a total of 1771 women with a positive cone margin, recurrence was identified in 272 of 1568 unvaccinated women. In comparison, reductions were observed in the 84 prophylactically vaccinated and 119 women vaccinated post-excision, with only 6 recurrence cases documented in each group. Henere stated that vaccination before treatment may lower the rate of post-treatment HSIL compared with non-vaccinated women (0.9% vs. 6.5%; *p* = 0.047) and, secondly, vaccination in women with persistent HPV infection after treatment might lower the prevalence of post-treatment HSIL (2.6% vs. 10.5%; *p* = 0.043). Finally, Krog L. [13] showed that vaccination before 15 years had a 35% lower risk for progression to CIN3 or worse (adjusted relative risk, 0.65; 95% confidence interval, 0.57–0.75), whereas vaccination between 15 and 20 years had a 14% lower risk (adjusted relative risk, 0.86; 95% confidence interval, 0.79–0.95). In contrast, for those vaccinated after the age of 20, the risk was comparable to that among women who were not vaccinated (adjusted relative risk, 1.02; 95% confidence interval, 0.96–1.09) [10].

## 4. Discussion

This summary aimed to review the literature on HPV vaccination with the 9-valent vaccine in individuals with HPV infection or histopathologically confirmed HPV-related precancerous lesions of the cervix. Despite the introduction of preventive vaccinations into the market and clinical practice, we will still need to wait a dozen or so years to observe the generation of adult women at the age at which patients currently most often suffer from HPV-related diseases. Furthermore, the vaccination rate among both adults and children in many countries remains insufficient to achieve herd immunity. The review included European countries with different oncology strategies and vaccination policies. In 2017, the public health system of Catalonia, Spain, started administering the HPV vaccine for women undergoing treatment for SIL, including those treated within the previous year (from July 2016). The first dose of the vaccine was administered either immediately before or after the treatment.

In Italy, a vaccination program against HPV started in 2006, but Gardasil 9 was adopted in 2017. According to the Italian National Immunisation Plan, catch-up vaccination is recommended for women up to 26 years of age and for men up to 18 years of age. The Italian government also recommends vaccination against HPV for women treated for CIN2+ or higher-grade injuries and for individuals with HIV infections. In Poland and the Czech Republic, as of 28 July 2025, adult vaccination is not publicly reimbursed, and patients wishing to receive the vaccination must cover the full cost themselves. As noted, the HPV vaccine is primarily recommended for young people as part of the national childhood vaccination program, usually starting at age 12. However, it can also be administered to adults up to age 45, especially those who have not previously received the vaccine, including individuals in at-risk groups. The vaccine is available for both males and females, and is fully covered.

The studies presented in the review not only differed in their vaccination strategies but were also heterogeneous in terms of comparable endpoints and outcomes.

It should be noted that although the follow-up time was similar in the studies, the timing of HPV vaccination administration in relation to the procedure significantly differed—before surgery, on the day of cervical conization, or after the excisional procedure. These differences may have key impacts on the expected effect of HPV disappearance and reduced risk of CIN 2+ recurrence after the procedure. Remission of HPV in SIL has been previously observed: low-grade cervical lesions might disappear spontaneously with vaccination, especially in young patients, while progression towards HSIL is more common in older women. LEEP without additional vaccination may have contributed to the regression of the infection in nearly 65% of patients [17,18]. We believe that the time of the first dose administration should be given as soon as possible. It may be provided at the time of HPV infection, at the time of diagnosis of SIL, before starting treatment for HSIL-CIN 2 and CIN 3 lesions, or after their completion. The most commonly used schedule for the 9-valent vaccine is the 0–2–6 schedule. The literature continues to report new cancer therapies, including the prevention of recurrence of HPV-related cervical diseases, for example, through O_3_-O_2_ vaginal insufflation. This represents an interesting and worthwhile future application of this technology, as the procedure is relatively minimally invasive [19]. Monitoring chronic HPV infection in vaccinated and unvaccinated individuals will require the use of the latest molecular diagnostic tools. These include tests detecting HPV E6/E7 mRNA and the methylation test [20]. Further development of this technology may provide patients with personalized therapeutic vaccines in the future. The results support consideration of additional HPV vaccination in HPV-positive adult women, but definitive conclusions require well-designed, prospective trials with standardized outcome measures.

Our study has several limitations. Firstly, most of the available evidence came from retrospective cohort studies; only two studies were prospective studies. Secondly, there was considerable heterogeneity in the study designs and follow-up definitions. Finally, all presented studies were conducted in Europe, with a limited population from other regions.

Based on the evidence synthesized in this review, we can propose several practical recommendations. Adult women who are HPV-positive or have histologically confirmed SIL should be offered the 9-valent HPV vaccine, particularly when undergoing excisional procedures. The most significant benefit appears when the first vaccine dose is administered before surgical treatment. To strengthen the evidence base, prospective, randomized, multicenter trials with standardized outcome definitions are required. Such studies would help confirm the observed benefits and guide consistent global recommendations.

## 5. Conclusions

Our analysis shows the effectiveness of vaccination against HPV in HPV-positive adult women and those with histopathological confirmed SIL. The greatest effect is observed when the first dose is given before the excision procedure. Although evidence is limited by retrospective designs and heterogeneity, the findings support extending vaccination to adult women and highlight the need for standardized guidelines and prospective studies. Further observation in 10 to 20 years should confirm the effects of vaccination on the reduction in HPV-related conditions.

## Figures and Tables

**Figure 1 viruses-17-01377-f001:**
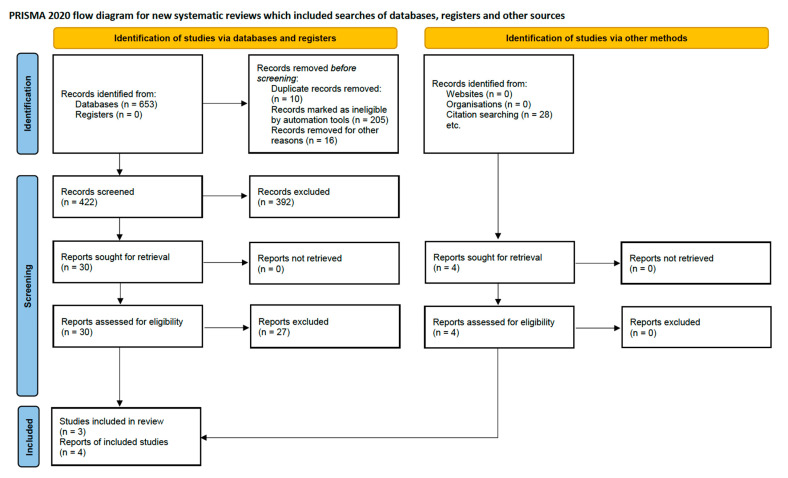
PRISMA flowchart.

**Figure 2 viruses-17-01377-f002:**
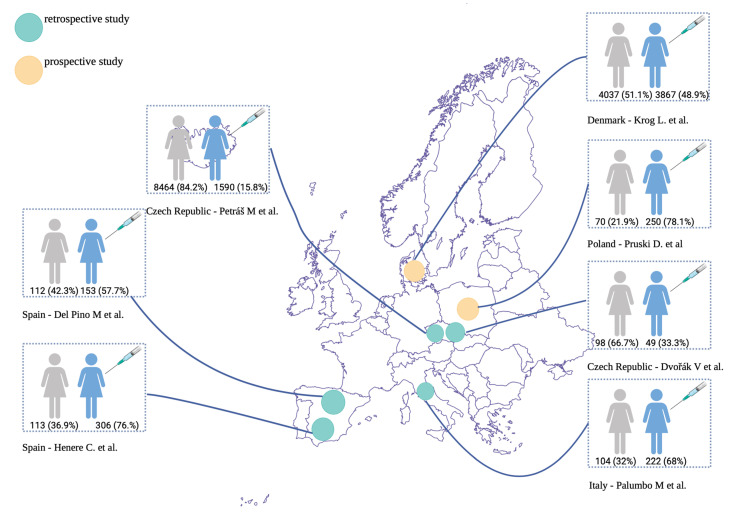
Division of retrieved studies into prospective and retrospective studies [9,10,11,12,13,14,15].

**Table 1 viruses-17-01377-t001:** Search strategy.

Database	Number of Results	Search Strategy
PubMed	393	(9-valent vaccine) OR (Gardasil9) AND ((IN) AND (HPV positive) OR (SIL) OR (squamous intraepithelial lesion) OR (HPV infection) OR (conization) OR (after treatment) OR (LEEP) OR (LLETZ))
Scopus	145	(“HPV vaccination” OR “9-valent vaccination” OR “Gardasil 9”) AND (“HPV positive” OR “SIL” OR “HSIL” OR “after treatment”) AND “women”
Cochrane	115	9-valent vaccine in HPV-positive patients → 1 HPV vaccination after LEEP → 20 HPV vaccination in HPV-positive patients → 77 HPV vaccination after LLETZ → 6 HPV vaccination after conization → 11 9valent vaccine after LLETZ → 0

**Table 2 viruses-17-01377-t002:** Inclusion and exclusion criteria.

Inclusion Criteria	Exclusion Criteria
1. Studies on the impact of vaccinating HPV-positive subjects with the 9-valent vaccine	1. Studies on other topics (e.g., the impact of a prophylactic vaccine, vaccines other than the 9-valent vaccine, or HPV-negative populations)
2. Vaccination of only adult women (≥18 years)	2. Vaccination in children, men
3. Only the HPV-positive population or those with squamous intraepithelial lesions of the cervix	3. Healthy population
4. Only finished studies	4. Unfinished studies
5. Studies in the English language	5. Studies in a language other than English
6. Only original studies with full-text access online	6. Case reports, reviews, meta-analyses, book chapters, conference presentations, letters, and original works without online full-text access

**Table 3 viruses-17-01377-t003:** Newcastle–Ottawa quality assessment scale (NOS) and description of studies.

Study	Dvořák V. et al. (2024) [9]	Palumbo M. et al. (2025) [10]	Petráš M. et al. (2025) [11]	Del Pino M. et al. (2020) [12]	Krog L. et al. (2024) [13]	Pruski D. et al. (2025) [14]	Henere C. et al. (2022) [15]
Main Aims of the Study	To evaluate the impact of post-excisional administration of the 9-valent HPV vaccine on the risk of CIN2+ recurrence in women.	To evaluate the effect of adjuvant 9-valent HPV vaccination following surgical excision or ablation in women with persistent low-grade (CIN1) or high-grade (CIN2–3) cervical intraepithelial neoplasia, compared with excision or ablation alone.	To estimate the effect of HPV vaccination on CIN2+ recurrence in relation to the timing of vaccination, administered either before or after cervical excisional treatment.	To compare the risk of persistent or recurrent HSIL following cervical conization between HPV-vaccinated and unvaccinated women.	To investigate whether HPV vaccination reduces the risk of progression from CIN2 to CIN3 or worse among women undergoing active surveillance, compared with unvaccinated women.	To evaluate cytological and HPV DNA outcomes following a full course of 9-valent HPV vaccination in women with HPV infection detected on cervical smear.	To evaluate the effect of vaccination timing (before or after excisional treatment) on protection against HSIL, as well as to assess the impact of vaccination on reducing post-treatment lesions in patients with persistent HPV infection following HSIL excisional treatment.
Study Design	Retrospective cohort study	Single-center retrospective observational study	Retrospective cohort study	Retrospective cohort study	Population-based cohort study	Prospective, ongoing, non-randomized study	Retrospective cohort study
Population (study group and control group)	98 unvaccinated and 49 vaccinated women.	The vaccinated group comprised 68% (222/326) of participants, while 32% (104/326) were unvaccinated.	Of the 10,054 women enrolled, 919 were vaccinated after conization, 502 prophylactically, and 169 had an undetermined timing of vaccination.	265 women were included in the study; 153 women (57.7%) accepted vaccination, and 112 (42.3%) refused the vaccine.	7904 women, of whom 3867 (48.9%) were vaccinated at least 1 year before a diagnosis of CIN2.	Of 320 patients, 250 (78.1%) decided to be vaccinated, and 70 (21.9%) did not.	Vaccinated group: 306 (76.9%), of which 113 (36.9%) had the first dose before excision and 193 (63.1%) after; unvaccinated group: 92 (23.1%).
Time of Vaccination	On the day of conization or later, after surgical excision.	The first dose was administered either before surgery or within 30 days post-surgery.	Women were classified as vaccinated prophylactically or post-excision if their last HPV vaccine dose was administered before or within one year of conization, respectively.	The first dose of the vaccine was scheduled after HSIL/CIN2-3 diagnosis, and it was provided either immediately before or after conization; women who had conization from July 2016 to July 2017 and had not been previously vaccinated had the first dose within 1–12 months after HSIL/CIN2-3 diagnosis.	Exposure was defined as having received ≥1 dose of a human papillomavirus vaccine at least 1 year before the CIN2 diagnosis.	The first dose before conization or a month after conization.	113 with the first dose before excision and 193 with the first dose after excision.
Results	The incidence rate of repeat conization: 18 per 100,000 person-days in the unvaccinated cohort, and 2 per 100,000 person-days in the vaccinated group.	Among women treated for CIN1, a positive HPV test was observed in 38% of unvaccinated women versus 18% of vaccinated women (*p* = 0.0169). In women treated for CIN2–3, 18% of unvaccinated women tested positive for HPV, compared with 8% in the vaccinated group (*p* = 0.0353).	CIN2+ recurrence was substantially reduced with vaccination. Among women with positive cone margins, 272 of 1568 unvaccinated women (51.6 per 1000 person-years) experienced recurrence, compared with only 6 cases each among 84 prophylactically vaccinated and 119 post-excision vaccinated women. Recurrence within 6 months of conization was reduced by 80% with prophylactic vaccination and 89% with post-excision vaccination.	Persistent/recurrent HSIL was less frequent in vaccinated than in unvaccinated women (3.3% vs. 10.7%, *p* = 0.015). HPV vaccination was associated with a reduced risk of persistent/recurrent HSIL. Vaccine uptake increased significantly, from 35.9% to 79.1%, when publicly funded.	Vaccination before age 15 reduced the risk of progression to CIN3 or worse by 35%, vaccination between ages 15 and 20 reduced the risk by 14%, while vaccination after age 20 showed no significant effect compared with unvaccinated women.	Post-vaccination outcomes compared with controls were as follows: same genotypes, 6.4% vs. 32.9%; partially same genotypes, 5.2% vs. 8.6%; complete remission, 72.4% vs. 45.7%; new infection, 12.4% vs. 5.7%; same/partially same genotype plus new infection, 3.6% vs. 7.1%.	Vaccination before treatment was associated with a lower rate of post-treatment HSIL compared with non-vaccinated women (0.9% vs. 6.5%; *p* = 0.047). Among women with persistent HPV infection after treatment, vaccinated women also had a lower prevalence of post-treatment HSIL than non-vaccinated women (2.6% vs. 10.5%; *p* = 0.043).
Conclusions	Post-conization HPV vaccination reduced the risk of recurrence of high-grade lesions by 87% (95% CI: 19–100%).	The human papillomavirus 9-valent vaccine was associated with a significant reduction in the proportion of women with a positive HPV test.	Regardless of timing, HPV vaccination has a beneficial long-term effect in lowering the risk of CIN2+ recurrence. The greatest benefit was observed in the first 6 months post-conization, with a positive cone margin.	HPV vaccination in women undergoing conization is associated with a 4.5-fold reduction in the risk of persistent/recurrent HSIL. Vaccination policies have an important impact on vaccination compliance.	Women who were vaccinated and who were undergoing active surveillance for CIN2 had a lower risk for progression to CIN3 or worse during 28 months of follow-up, when compared with women who were not vaccinated, but only if the vaccine was administered before the age of 20 years.	Vaccination against HPV significantly affects the disappearance of the viral infection in women not vaccinated during puberty. A statistically significant disappearance of HPV infection occurs in patients, both those diagnosed with HPV and undergoing LEEP.	HPV vaccination before treatment reduces the prevalence of post-treatment HSIL, suggesting that vaccination might even benefit women with persistent HPV after treatment.
Timeframe	Jan 2014 to Aug 2023	2020–2024	2010–2024	Jan 2013 to Jul 2018	Jan 2007 to Dec 2020	2020–2023	Jul 2016 to Dec 2019
Country	Czech Republic	Italy	Czech Republic	Spain	Denmark	Poland	Spain
National Vaccination Program	2012, for 13-year-old girls; 2015, nonavalent vaccination for 13-year-old girls; 2024, nonavalent vaccination for 11–14-yo boys and girls.		2012, for 13-year-old girls; 2015, nonavalent vaccination for 13-year-old girls; 2024, nonavalent vaccination for 11–14-year-old boys and girls.	In July 2017, the public health system of Catalonia started funding the HPV vaccine for women undergoing treatment.			In July 2017, the public health system of Catalonia started funding the HPV vaccine for women undergoing treatment.
Grade	Very good	Very good	Very good	Very good	Very good	Very good	Very good

**Table 4 viruses-17-01377-t004:** Main outcomes in vaccinated and unvaccinated groups.

Outcome	Vaccinated Group	Unvaccinated Group	*p*-Value
HPV complete remission	72.4%	45.7%	
CIN2+ recurrence after conization	3.3%	10.7%	0.015
Progression to CIN3	0.9%	6.5%	
Positive HPV result in LSIL	18%	38%	0.0169
Positive HPV result in HSIL	8%	18%	0.0353

## Data Availability

Not applicable.

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
