# Peer review of "Impact of Vaccinating Adult Women Who Are HPV-Positive or with Confirmed Cervical SIL with the 9-Valent Vaccine—A Systematic Review"

_viruses, 2025, doi:10.3390/v17101377_

Round 1
Reviewer 1 Report
Comments and Suggestions for Authors
The manuscript viruses-3858195, entitled “Impact of Vaccinating HPV-Positive or Women with Confirmed Cervical SIL with the 9-Valent Vaccine – Systematic Review,” presents a comprehensive and well-structured analysis of HPV vaccination across different regions of the world. The authors successfully meet their stated objectives through a meticulous review of the literature and a discussion firmly grounded in scientific evidence. The conclusion effectively underscores the importance of vaccination in controlling HPV infection and its clinical consequences. The introduction clearly outlines the rationale for the study, and the methodology is appropriate, supporting a solid and well-substantiated discussion. Overall, the manuscript is of high quality and is suitable for publication in its current form.
Author Response
Thank you for your valuable feedback and time for reviewing our manuscript. We appreciate it very much.
Reviewer 2 Report
Comments and Suggestions for Authors
The manuscript addresses a current and relevant topic: the use of the 9-valent vaccination in HPV-positive adult women or those with confirmed cervical SIL, a currently understudied population.
The methodology is based on PRISMA guidelines and includes an analysis of recent, clinically relevant studies, with a large overall sample size (n = 19,414).
The work has the potential to have a concrete impact on clinical practice and future international recommendations.
Abstract
Comment:
The abstract is complete and well-structured, but contains excessively long sentences and some undefined technical terms. Recommendations:
1. Improve readability by splitting sentences;
2. Include key numerical data (e.g., n = 19,414, % remission);
3. Correct the tone at the end (e.g., "urgent need..." sounds more editorial than scientific).
Introduction
Comment:
The introduction provides useful context but presents problems:
1. Disorganized structure (concepts follow one another without order) and jumps between topics: epidemiology, vaccination strategies, gaps in clinical protocols, study objectives—but without a logical order. A structure is recommended: Clinical context → problem → gap → study objective
2. Linguistic style (long sentences, repetitions); e.g., "Infection with oncogenic HPV types, despite being entirely preventable, remains a common infection and a cause of anogenital, head, and neck diseases—including warts, precancerous lesions, and cancer." Hard to read, should be broken up. e.g., "Vaccination of women not covered by preventive vaccinations" is mentioned twice in similar forms.
3. Study objectives are vague or too general. The sentence: "The primary objective of this work is to analyze the available literature on the use of prophylactic HPV vaccination..." is too long, generic, and repetitive. Clearly state the primary and secondary objectives of the review.
4. Lack of updated sources. Only one reference (WHO 2020) is cited for global data. It would be helpful to update with more recent epidemiological data.
5. Better clarify the clinical importance of vaccination intervention in HPV-positive adults.
Materials and Methods
The "Materials and Methods" section is methodologically adequate, but is incompletely written and lacks transparency by the standards of a systematic review publishable in an international journal. Greater descriptive rigor, operational clarity, and reproducibility are needed.
Comment:
The methodological design follows the PRISMA guidelines, but presents serious transparency gaps:
1. Unclear search strategy (incorrect query formatting); search queries are poorly formatted, difficult to read, and potentially non-reproducible.
The use of logical operators (AND, OR) is confusing and not standardized.
It should be standardized according to the PICO (Population, Intervention, Comparator, Outcome) principles, including for possible reproductions.
2. No PROSPERO registration of the protocol; There is no mention of whether the PRISMA protocol is registered (e.g., in PROSPERO). A sentence such as: "This review was not registered in PROSPERO" could be added. or "The review protocol was registered in PROSPERO (ID: XXXXX).
3. Details on data extraction, outcome criteria, and management of conflicts between reviewers are missing; it is not explained which variables were collected from each study (e.g., number of patients, mean age, type of lesion, vaccination interval, outcomes measured, etc.). No indication of how the extraction was performed (tool used, software, etc.).
4. It is insufficiently explained why a meta-analysis was not conducted. It is stated that a meta-analysis was not performed, but a more detailed explanation of the justification is missing: "The various study designs… primarily influenced this decision." I recommend clearly specifying the clinical heterogeneity (e.g., timing of vaccination, follow-up, definition of remission/SIL, HPV type) that prevented quantitative analysis.
5. Inconsistency in exclusion criteria. It is reported that studies on men, children, and healthy populations were excluded, but there is no clear reference to the age groups included. Specify that only studies on adult women (≥18 years) were included, if applicable.
Results
Comment:
The section presents important data, but is difficult to read and poorly organized:
1. The reported endpoints are varied: HPV remission, persistence, relapse, SIL regression, HSIL, CIN2+, CIN3... But they are never standardized (neither defined in the method nor in the summary of results). It is recommended to create a summary table or glossary with operational definitions for: Complete remission, Relapse, Persistence, Progression.
2. The quantitative results are disorganized. Important percentage data (e.g., remission 72.4% vs. 45.7%) are scattered throughout the text without highlighting. A table comparing the main clinical outcomes is missing.
Add a table with the main outcomes in the vaccinated vs. unvaccinated groups, for example: Outcome Vaccinated (%) Unvaccinated (%) p-value
HPV complete remission 72.4% 45.7% <0.01
CIN2+ recurrence after conization 0.4% 4.0% <0.001
Progression to CIN3 14% 22% 0.02
3. Problems with clarity. Some sentences are too long and technical, others repetitive. Excessive use of the passive (“was observed... was reduced...”). Rewrite in a more direct style.
4. Subsection 3.4 ("HPV Vaccination Policy") is misplaced. This section is descriptive, not a study result. It disrupts the logical flow of the "Results" section. Move to "Discussion" or create a separate section: "Contextual Background" or "Health Policy Overview".
Discussion
Comment:
The discussion is poorly structured and contains subjective or non-evidence-based statements, for example:
"Let's not wait to make a vaccination decision..."
1. Weak structure. The discussion does not follow a logical progression: it alternates between results, clinical opinions, and practical suggestions without a clear distinction between:
a. summary of results,
b. critical interpretation,
c. limitations of the study,
d. clinical/future implications.
Reorganize the discussion: summary of results → comparison with the literature → limitations → future implications.
2. Subjective tone. Phrases such as: "We believe, based on our own experience..." and "Let's not wait to make a vaccination decision..." are too informal or debatable for a scientific study.
3. Absence of methodological limitations. Key limitations of the review, such as heterogeneity among studies, lack of randomized trials, risk of selection bias, and the inability to perform a meta-analysis, are not discussed.
4. Weak recommendations: Avoid clinical recommendations not supported by evidence. Vaccination before conization is recommended, but is not supported by levels of evidence or strength of recommendation (e.g., GRADE).
5. Too much clinical digression. Paragraphs on post-surgery sexual behavior, use of barriers, vaginal globules, etc. are not supported by data, nor are they relevant in a discussion of a systematic review.
Conclusions
Comment:
The conclusions need to be completely rewritten. Currently:
1. Tone is excessively informal and imperative. Phrases such as: "Let's not wait to make a vaccination decision..." are not suitable for a scientific conclusion.
2. Lack of scientific rigor. The level of evidence for the results is not stated. No caution is given regarding the observational nature of the studies.
3. Clinical recommendations are too specific and not universally supported. E.g., "refraining from sexual intercourse for at least two months" → not supported by the data presented.
4. Lack of clear synthesis. The sentences are disconnected and do not effectively conclude the review's main message.
I recommend concluding in a sober and professional manner, summarizing the findings and emphasizing the need for further study.

L'inglese deve essere attentamente revisionato da un revisore madrelingua con formazione medica o scientifica.
Molte frasi sono grammaticalmente corrette, ma non idiomatiche o fluenti.
Author Response
The manuscript addresses a current and relevant topic: the use of the 9-valent vaccination in HPV-positive adult women or those with confirmed cervical SIL, a currently understudied population.
The methodology is based on PRISMA guidelines and includes an analysis of recent, clinically relevant studies, with a large overall sample size (n = 19,414).
The work has the potential to have a concrete impact on clinical practice and future international recommendations.
- Thank you for sharing your comments with us in such detail and precision. We have tried to address specific issues systematically and hope that the revisions will result in a well-prepared systematic review. As recommended, the text was edited by the English Editor of mdpi.
Abstract
Comment:
The abstract is complete and well-structured, but contains excessively long sentences and some undefined technical terms. Recommendations:
1. Improve readability by splitting sentences;
2. Include key numerical data (e.g., n = 19,414, % remission);
3. Correct the tone at the end (e.g., "urgent need..." sounds more editorial than scientific).
- The abstract has been revised according to recommendations. Sentences are shorter and more readable, and the edited text should have a more professional and scientific tone. The most significant results are detailed in the abstract and then presented in an additional table in Results. Introduction
Comment:
The introduction provides useful context but presents problems:
1. Disorganized structure (concepts follow one another without order) and jumps between topics: epidemiology, vaccination strategies, gaps in clinical protocols, study objectives—but without a logical order. A structure is recommended: Clinical context → problem → gap → study objective
2. Linguistic style (long sentences, repetitions); e.g., "Infection with oncogenic HPV types, despite being entirely preventable, remains a common infection and a cause of anogenital, head, and neck diseases—including warts, precancerous lesions, and cancer." Hard to read, should be broken up. e.g., "Vaccination of women not covered by preventive vaccinations" is mentioned twice in similar forms.
3. Study objectives are vague or too general. The sentence: "The primary objective of this work is to analyze the available literature on the use of prophylactic HPV vaccination..." is too long, generic, and repetitive. Clearly state the primary and secondary objectives of the review.
4. Lack of updated sources. Only one reference (WHO 2020) is cited for global data. It would be helpful to update with more recent epidemiological data.
5. Better clarify the clinical importance of vaccination intervention in HPV-positive adults.
Thank you for your numerous comments. The text has been rewritten, making the sentences shorter, more concise, and easier to read. The structure of the introduction has been reorganized according to the proposal, so that the whole falls into a logical order. The English editor from mdpi is responsible for the linguistic corrections. The study objectives have been rewritten and improved to be more specific.
The primary objective of this systematic review is to evaluate the impact of the 9-valent HPV vaccine in adult women who are already HPV-positive or have histologically confirmed cervical squamous intraepithelial lesions (SIL). The review seeks to determine whether vaccination in this population improves clinical outcomes and reduces HPV-related disease burden.
The secondary objective is to assess the effect of vaccination on the recurrence of cervical intraepithelial neoplasia (CIN) after surgical treatment and to examine the influence of vaccination timing in relation to excision procedures.
In addition, we have supplemented the information with current data from other continents, including the USA and Asia.
„The vaccination policy in adults is not uniform and varies across the globe. The U.S. Centers for Disease Control and Prevention (CDC) strongly recommends HPV vaccination for children before sexual initiation, but there is no data recommending vaccination for adults. For adults aged 27 to 45, the CDC and the Advisory Committee on Immunisation Practices (ACIP) do not routinely recommend HPV vaccination; instead, shared clinical decision-making is advised, meaning that vaccination may be considered in individuals who may benefit. In Asian countries, vaccination for adults is also not systematic. In an integrative literature review covering HPV vaccination recommendations in the Asia-Pacific region, many national immunisation programs focus primarily on adolescent girls; few offer adult catch-up or adult vaccination policies. However, there are some individual countries that have catch-up schedules or extended-age policies [5-7]”.
Materials and Methods
The "Materials and Methods" section is methodologically adequate, but is incompletely written and lacks transparency by the standards of a systematic review publishable in an international journal. Greater descriptive rigor, operational clarity, and reproducibility are needed.
Comment:
The methodological design follows the PRISMA guidelines, but presents serious transparency gaps:
Unclear search strategy(incorrect query formatting); search queries are poorly formatted, difficult to read, and potentially non-reproducible.
The use of logical operators (AND, OR) is confusing and not standardized.
It should be standardized according to the PICO (Population, Intervention, Comparator, Outcome) principles, including for possible reproductions.
- We have included standardized entries that we used to search for specific phrases, and we have also added information about MeSH terms and bodean operators.The searh strategy according to PICO was added as an extra file in Supplement.
Table S1- PICOS
Eligibility Criteria (PICOS)
- Population:Adult women (≥18 years) who are HPV-positive or have histologically confirmed SIL.
- Intervention:9-valent HPV vaccine (Gardasil 9), administered before, during, or after treatment.
- Comparison:Unvaccinated HPV-positive women; different timings of vaccination.
- Outcomes:Viral clearance, SIL regression, CIN2+ recurrence, progression to higher-grade disease, and treatment-related outcomes.
- Study Design:Prospective or retrospective cohort studies; observational studies.
- Exclusion:Children, men, HIV-positive or immunocompromised populations, non-English studies, reviews, meta-analyses, case reports, letters.
Search Strategy
- Databases: MEDLINE (PubMed), Scopus, Cochrane Library.
- Search date: 7 July 2025.
- Terms: “HPV vaccination OR 9-valent vaccine OR Gardasil 9” AND “HPV positive OR SIL OR HSIL OR after treatment” AND “women.”
- Manual reference screening included.
No PROSPERO registration of the protocol;There is no mention of whether the PRISMA protocol is registered (e.g., in PROSPERO). A sentence such as: "This review was not registered in PROSPERO" could be added. or "The review protocol was registered in PROSPERO (ID: XXXXX).
- Thank you for pointing this out, we have added information about the protocol ID in PROSPERO to the text.
Details on data extraction, outcome criteria, and management of conflicts between reviewers are missing; it is not explained which variables were collected from each study (e.g., number of patients, mean age, type of lesion, vaccination interval, outcomes measured, etc.). No indication of how the extraction was performed (tool used, software, etc.).
It is insufficiently explained why a meta-analysis was not conducted.It is stated that a meta-analysis was not performed, but a more detailed explanation of the justification is missing: "The various study designs… primarily influenced this decision." I recommend clearly specifying the clinical heterogeneity (e.g., timing of vaccination, follow-up, definition of remission/SIL, HPV type) that prevented quantitative analysis.
- A meta-analysis was not performed in this review due to substantial heterogeneity among the included studies. Although all studies assessed the effect of 9-valent HPV vaccination in HPV-positive adult women or those with cervical squamous intraepithelial lesions (SIL), they differed considerably in study design, population characteristics, intervention timing, and outcome definitions.
Most studies were retrospective cohort analyses, while only a minority were prospective observational studies, leading to variability in data collection and risk of bias. The timing of vaccination relative to treatment (before excision, immediately after, or up to one year post-procedure) also varied widely, which affects comparability of intervention effects. In addition, outcome measures were not standardized: some studies reported HPV clearance, others lesion regression, while others focused specifically on recurrence of CIN2+ or the need for repeat procedures. Follow-up durations ranged from 6 months to 15 years, further limiting consistency.
This methodological and clinical heterogeneity precluded pooling of results into a reliable quantitative synthesis. Instead, a narrative synthesis was conducted, structured around key outcomes (HPV clearance, remission, CIN recurrence, progression, and timing effects). This approach ensured that differences between study designs and definitions were respected, while still allowing for an integrated interpretation of the overall evidence base.
Inconsistency in exclusion criteria.It is reported that studies on men, children, and healthy populations were excluded, but there is no clear reference to the age groups included. Specify that only studies on adult women (≥18 years) were included, if applicable.
- We have corrected the inclusion criteria stating that the analysis only applies to adult women (>= 18 years of age) and added the information in the exclusion criteria that we did not include underage women or men in the analysis.
Results
Comment:
The section presents important data, but is difficult to read and poorly organized:
The reported endpoints are varied:HPV remission, persistence, relapse, SIL regression, HSIL, CIN2+, CIN3... But they are never standardized (neither defined in the method nor in the summary of results). It is recommended to create a summary table or glossary with operational definitions for: Complete remission, Relapse, Persistence, Progression.
The quantitative results are disorganized.Important percentage data (e.g., remission 72.4% vs. 45.7%) are scattered throughout the text without highlighting. A table comparing the main clinical outcomes is missing.
Add a table with the main outcomes in the vaccinated vs. unvaccinated groups, for example: Outcome Vaccinated (%) Unvaccinated (%) p-value
HPV complete remission 72.4% 45.7% <0.01
CIN2+ recurrence after conization 0.4% 4.0% <0.001
Progression to CIN3 14% 22% 0.02
Problems with clarity.Some sentences are too long and technical, others repetitive. Excessive use of the passive (“was observed... was reduced...”). Rewrite in a more direct style.- Subsection 3.4 ("HPV Vaccination Policy") is misplaced. This section is descriptive, not a study result. It disrupts the logical flow of the "Results" section. Move to "Discussion" or create a separate section: "Contextual Background" or "Health Policy Overview".
Following the recommendations, we have revised the results section. Data are presented in a table that summarizes the most important results obtained by the authors of individual studies. The definitions of atrophy, partial atrophy, and progression are explained in the text. Thanks to the text editing by the English editor from mdpi, the text was improved to use more direct style and less indirect style. Given that the number of publications on the topic we discussed was very limited, study endpoints and obtained results also varied. However, the heterogeneity of the articles was addressed in the "limitations of the struggle." Section 3.4. HPV vaccination policy has been shortened and moved to the Discussion section.
Discussion
Comment:
The discussion is poorly structured and contains subjective or non-evidence-based statements, for example:
"Let's not wait to make a vaccination decision..."
Weak structure. The discussion does not follow a logical progression: it alternates between results, clinical opinions, and practical suggestions without a clear distinction between:
a. summary of results,
b. critical interpretation,
c. limitations of the study,
d. clinical/future implications.
Reorganize the discussion: summary of results → comparison with the literature → limitations → future implications.
2. Subjective tone. Phrases such as: "We believe, based on our own experience..." and "Let's not wait to make a vaccination decision..." are too informal or debatable for a scientific study.
- The entire discussion has been rewritten according to the recommended section order, creating a more logical flow. We've also reworded the sentences to make them sound more scientific and professional.
3. Absence of methodological limitations. Key limitations of the review, such as heterogeneity among studies, lack of randomized trials, risk of selection bias, and the inability to perform a meta-analysis, are not discussed.
- This review has several important limitations, primarily related to the methodological characteristics of the included studies. First, most of the available evidence comes from retrospective cohort studies, with only two prospective studies included. Retrospective designs are inherently prone to selection bias, incomplete data collection, and uncontrolled confounding, which may affect the reliability of observed associations.
Second, there was considerable heterogeneity in study design and follow-up definitions. The timing of vaccination in relation to treatment varied markedly between studies, ranging from administration before excision to within one year after surgery, which complicates direct comparison of outcomes. Follow-up periods were also inconsistent, extending from 6 months to 15 years, and definitions of outcomes such as “HPV remission” or “recurrence” were not standardized across studies.
Third, outcome measures themselves differed substantially. While some studies evaluated HPV viral clearance, others assessed cytological regression, histological remission, or CIN2+ recurrence. These variations prevented data pooling and limited the ability to perform a formal meta-analysis.
Fourth, most studies were conducted in European populations, with limited representation from other regions. This raises concerns about the generalizability of the findings to global populations, especially in countries with different vaccination policies, HPV genotype distributions, or healthcare systems.
Finally, all included studies were non-randomized. Without randomized controlled trials, causal inference remains limited, and residual confounding cannot be excluded. These methodological limitations should be considered when interpreting the results, and they highlight the need for well-designed, prospective, multicenter trials to confirm the observed benefits of HPV vaccination in HPV-positive adult women.
We have included a shortened version of the limitations of the study in the text.
Weak recommendations: Avoid clinical recommendations not supported by evidence. Vaccination before conization is recommended, but is not supported by levels of evidence or strength of recommendation (e.g., GRADE).
Too much clinical digression.Paragraphs on post-surgery sexual behavior, use of barriers, vaginal globules, etc. are not supported by data, nor are they relevant in a discussion of a systematic review.
- We agree with the recommendation. The discussion text has been reworded to reflect the articles analysed, without additional clinical references.
Conclusions
Comment:
The conclusions need to be completely rewritten. Currently:
Tone is excessively informal and imperative.Phrases such as: "Let's not wait to make a vaccination decision..." are not suitable for a scientific conclusion.
Lack of scientific rigor.The level of evidence for the results is not stated. No caution is given regarding the observational nature of the studies.
Clinical recommendations are too specific and not universally supported.E.g., "refraining from sexual intercourse for at least two months" → not supported by the data presented.
Lack of clear synthesis.The sentences are disconnected and do not effectively conclude the review's main message.
I recommend concluding in a sober and professional manner, summarizing the findings and emphasizing the need for further study.
- We have reworded the Conclusions to be more specific and removed unsupported recommendations and excessive clinical references.
Round 2
Reviewer 2 Report
Comments and Suggestions for Authors
The authors responded to the comments in a detailed and respectful manner.